# Absolute and relative reliability of pain sensitivity and functional outcomes of the affected shoulder among women with pain after breast cancer treatment

G. H. F. Rasmussen[1]*, M. Kristiansen[1], M. Arroyo-Morales[2], M. Voigt[1], P. Madeleine[1]

**1** Department of Health Science and Technology, Sport Sciences - Performance and Technology, Aalborg University, Aalborg, Denmark, **2** Department of Physical Therapy, Faculty of Health Sciences, University of Granada, Granada, Spain

\* ghfr@hst.aau.dk

**Data Availability Statement:** All relevant data are within the paper and its Supporting Information files.

## Abstract

### Objective

Breast cancer survivors (BCS) are often characterized by decreased pressure pain thresholds (PPT), range of motion (ROM) and strength in and around the shoulder affected by the treatment. This intra-rater reliability study was to establish the relative and absolute reliability of PPT's, active ROM and maximal isokinetic muscle strength (MIMS) of the affected shoulder in BCS with persistent pain after treatment.

### Methods

Twenty-one BCS participated in the study. The PPTs of 17 locations and pain intensity were assessed using a pressure algometer and a numeric rating scale. The ROM was measured using a universal goniometer and MIMS was measured using an isokinetic dynamometer. Relative reliability was estimated using intra class correlation coefficient (ICC), and absolute reliability using standard error of measurement (SEM). Minimum detectable change (MDC) was calculated from SEM.

### Results

The ICCs for PPTs ranged from 0.88–0.97, with SEM values ranging from 12.0 to 28.2 kPa and MDC ranging from 33.2 to 78.2 kPa. The ICCs for ROM ranged from 0.66–0.97, with SEM values ranging from 3.0 to 7.5˚ and MDC ranging from 8.4 to 20.8˚. Finally, ICCs for MIMS ranged from 0.62–0.92, with SEM values ranging from 0.03 to 0.07 Nm/Kg FFM and MDC ranging from 0.09 to 0.19 Nm/kg FFM.

### Conclusion

The results of this study indicate that PPTs, ROM and MIMS can be measured reliably on the affected shoulder in BCS with pain after treatment. This offer the possibility of using these measures to assess the effectiveness of interventions in this population.

**Funding:** GHFR, PM, MK and MV received a grant (grant nr. R204-A12469) from the Danish Cancer Society (URL: https://www.cancer.dk/). The grant covered salary and equipment expenses. The funder had no role in study design, data collection and analysis, decision to publish, or preparation of the manuscript.

**Competing interests:** The authors have declared that no competing interests exist.

## Introduction

Breast cancer is the most common type of cancer among women. More than one million new cases were reported worldwide in 2018 [1]. Fortunately, due to new treatments and early detection, survival has improved over time [2] with a current 5-year survival rate of 90% [1]. However, breast cancer and its treatments result in both complications and common adverse effects, including severe pain and loss of shoulder function exemplified by limited shoulder range of motion (ROM) and decreased strength [3]. Pain in and around the surgical area affect 25–60% of the breast cancer survivors (BCS) [4], while 42–56% report difficulties in lifting the upper limb and 10–55% show restricted glenohumeral joint ROM [5]. These morbidities can be caused by scar tissue formation [6] and nerve damage [7] from the surgical incisions, and may be further exacerbated by soft tissue fibrosis secondary to radiation- and/or chemo therapy [8–10]. They may arise shortly after surgery and/or adjuvant treatment and can remain as a source of physical and psychological distress for several years [11]. Consequently, shoulder morbidity and pain after breast cancer treatment have received considerable attention in the literature with several recent reviews examining the extent of these issues [6], [12–14].

Importantly, pain after treatment for breast cancer can increase pain sensitivity [15] and reduce physical function [16]. Pain intensity is correlated with decreased muscle strength, ROM, job participation, and reduced use of the affected arm in leisure activities [17]. However, to the best of our knowledge, there are no studies linking shoulder pain sensitivity and physical function in BCS. Key methods for assessing pain sensitivity and shoulder function include pressure algometry, goniometry and isokinetic dynamometry. Pressure algometry can be used to assess mechanical pain sensitivity in different anatomical locations [18], and the assessment of multiple pressure pain threshold (PPT) locations enable the visualization and quantification of the spatial distribution of mechanical sensitivity to pain [19–21]. The spatial distribution of the mechanical pain sensitivity reveals the extent of hyperalgesia in BCS [22], [23], and localizing the most sensitive areas may be of interest to the design of specific rehabilitation programs [15]. Goniometric measurements can be performed with a universal goniometer, a low cost tool most commonly used for evaluating active and passive joint ROM in clinical settings [24]. Isokinetic dynamometry allows the measurement of maximum muscle strength with an accommodating resistance, at a constant angular velocity, thereby enabling maximum force production throughout a prescribed ROM [25].

The above mentioned methods have been used to assess mechanical pain sensitivity and shoulder function in BCS [15], [26], [27], and have been reported as reliable in both asymptomatic [25], [28], [29] and patient populations [30–32]. However, little is known about the relative and absolute reliability of these methods in BCS with pain after treatment. As previously established, ROM and muscle strength are affected by the presence of pain, and considering that substantial day-to-day fluctuations can be observed in perceived pain [33], measures of pressure algometry, goniometry and isokinetic dynamometry may vary considerably in BCS experiencing pain. In order to monitor the effectiveness of an intervention aimed at improving shoulder function and decreasing pain after breast cancer treatment, it is essential to have reliable tools for assessment of clinical outcomes [34]. Hence, the aim of the present study was to establish the spatial distribution of mechanical pain sensitivity, and the absolute and relative test-retest reliability of PPTs, active ROM and maximal isokinetic muscle strength (MIMS) in BCS with pain in body regions affected by the treatment, i.e. the ventral and dorsal regions of the shoulder.

## Methods

### Participants

According to the guidelines of Bujang & Baharum [35], assuming two observations per participant, an alpha level of 0.05, a beta level of 80% and an expected ICC of 0.60, the minimum sample size required for this study was 15 participants. To account for potential dropout it was decided to enrol 21 BCS. Participants were recruited by means of a database letter through the national database managed by the Danish Breast Cancer corporate Group. The inclusion and exclusion criteria were similar to those used by Caro-Morán & colleagues [15]. Accordingly, the inclusion criteria were: I) primary diagnosis of breast cancer (grades I-IIIA); II) adult women at least 18 years of age; III) having received breast cancer treatment (i.e. surgery and possible adjuvant chemo and/or radiotherapy) at least 18 months before the start of the study; IV) self reported pain in the areas of the breast, shoulder, axilla, arm and/or side of body with an intensity of $\geq 3$ on a numeric rating scale (0 = no pain, 10 = worst pain imaginable); V) no signs of cancer recurrence; VI) reading, writing and speaking Danish. The exclusion criteria were: I) breast surgery for cosmetic reasons or prophylactic mastectomy; II) bilateral breast cancer; III) lymphedema; IV) other chronic pain conditions (e.g., rheumatoid arthritis) or V) previous diagnosis of fibromyalgia syndrome. All the participants were instructed to maintain their normal everyday lifestyle, but avoid physical activity and consumption of alcohol, caffeine, nicotine, or painkillers in the last 24 hours prior to the experimental sessions. This was confirmed verbally upon arrival at the laboratory. The study protocol was approved by the local Ethics Committee (N-20180090) and conducted according to the Declaration of Helsinki. Following a detailed written and verbal explanation of the experimental risks of the study, the participants gave their written informed consent prior to participating in the study.

### Study design

This intra-rater reliability study was conducted in accordance with the GRRAS guidelines [36]. In line with the GRRAS guidelines, we report information on sample selection, study design, and statistical analysis (S1 Appendix). Each participant completed a familiarization session (FS), followed by two experimental sessions (ES1 & ES2), see Fig 1. In agreement with previous studies on PPT and MIMS in chronic pain populations [32], [37], familiarization and the two experimental sessions were separated by a time period of seven days. Baseline characteristics and profile on physical, health, surgical, medical and pain status were collected during

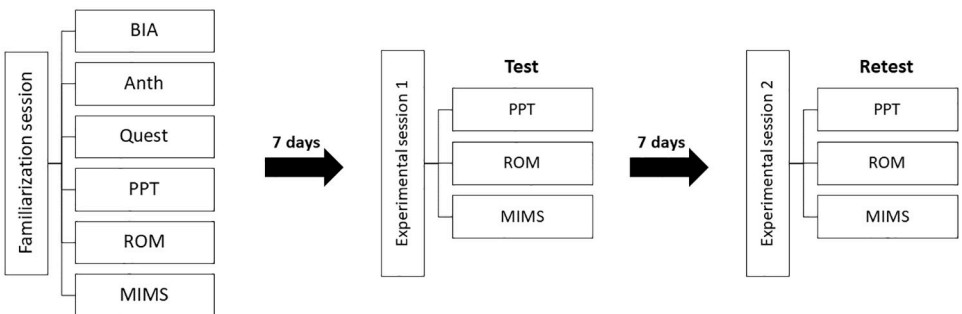

**Fig 1. Experimental overview.** Overview of the experimental protocol at test-retest. Abbreviations: Bioelectrical Impedance Analysis: BIA, Anthropometrics: Anth, Questionnaire: Quest, Pressure pain threshold: PPT, Active range of motion: ROM, Maximal isokinetic muscle strength: MIMS.

the FS. The PPT, active range of motion (ROM) and MIMS were measured unilaterally on the operated side in all sessions by the same experimenter (GHFR) (Fig 1).

## Outcomes

**Pain intensity and mechanical sensitivity measures.** *Pain intensity (PI)* during everyday living was rated for the chest, shoulder, axilla, arm and side of body at the beginning of FS using a 0–10 numeric pain rating scale [38]. The cut-off scores used as reference were: 0 = no pain; 1–3 = mild; 4–6 = moderate; 7–10 = severe pain [39]. The highest pain intensity rated across locations was reported as peak pain.

*Pressure pain thresholds* were measured unilaterally over 6 points on the dorsal and 11 points on the ventral part of the chest, shoulder, and neck regions for a total of 17 points. A single measurement was collected from the ipsilateral tibialis anterior muscle as a distant reference point [7]. For the measurements performed on the dorsal region, participants were seated with their chest supported and elbows resting on the knees with the humerus perpendicular to the floor. For measurements performed on the ventral region, participants were lying supine on an examination table with both arms resting along the body. All PPT measurements were made using a pressure algometer (Somedic AB, Farsta, Sweden) with a 1 cm$^2$ probe and a constant pressure rate of 30 kPa/s. The participants were instructed to press a handheld button as soon as the pressure sensation changed to pain sensation. The assessments were performed twice over two rounds in systematic order and a third time if the point assessed had a coefficient of variance over 20% [40]. There were approx. 6-minutes between measurements made over the same point to avoid temporal summation of pain.

Pressure pain threshold mapping of the dorsal region using the average PPT values of the six points was performed by measuring the distance *d* between C7 and acromion for each participant to compute the inter-distance between the six points covering the dorsal muscles (Fig 2). Similarly, PPT mapping of the ventral region was generated using the average PPT values of the 11 points and the distance *e* between acromion and the sternoclavicular joint for positioning of measuring points (Fig 3). Adjacent points were separated by varying distances according to their anatomical location relative to the reference points (Figs 2 and 3). We applied an inverse distance weighted interpolation to obtain a map of the spatial pressure pain distribution of the dorsal and ventral region [19]. For more details about the mathematical aspects, see Alburquerque-Sendin et al. [18]

**Physical measures.** *Body mass index (kg/m$^2$)* was calculated from height and weight measured at baseline. The fat free mass (FFM) was computed using bioelectrical impedance analysis (InBody 370, Biospace, Seoul, Korea) in agreement with the manufacturer's recommendations.

*Active ROM* was measured for supine shoulder flexion, supine horizontal shoulder flexion/ extension, supine internal/external shoulder rotation, and seated upright shoulder abduction (Fig 4A–4F). Similar levels of reliability have been reported for supine and upright measures of active ROM [28] and hence, positions were chosen to match the MIMS protocol. In agreement with Norkin & White [41], supine measurements were performed with the knees flexed to flatten the lumbar spine. For the seated upright measurement participants were positioned seated firmly against the back of the chair to ensure trunk stabilization, and instructed to maintain a neutral head position in agreement with Dougherty et al. [42].

Shoulder flexion ROM was measured as maximal shoulder flexion starting with the arm in 0˚-shoulder flexion, 0˚- abduction/adduction and rotation, 0˚ -elbow flexion and 0˚ -forearm supination/pronation (Fig 4A). Horizontal flexion/extension ROM was measured as maximal horizontal shoulder flexion/extension starting with the arm in 90˚ -shoulder flexion, 0˚

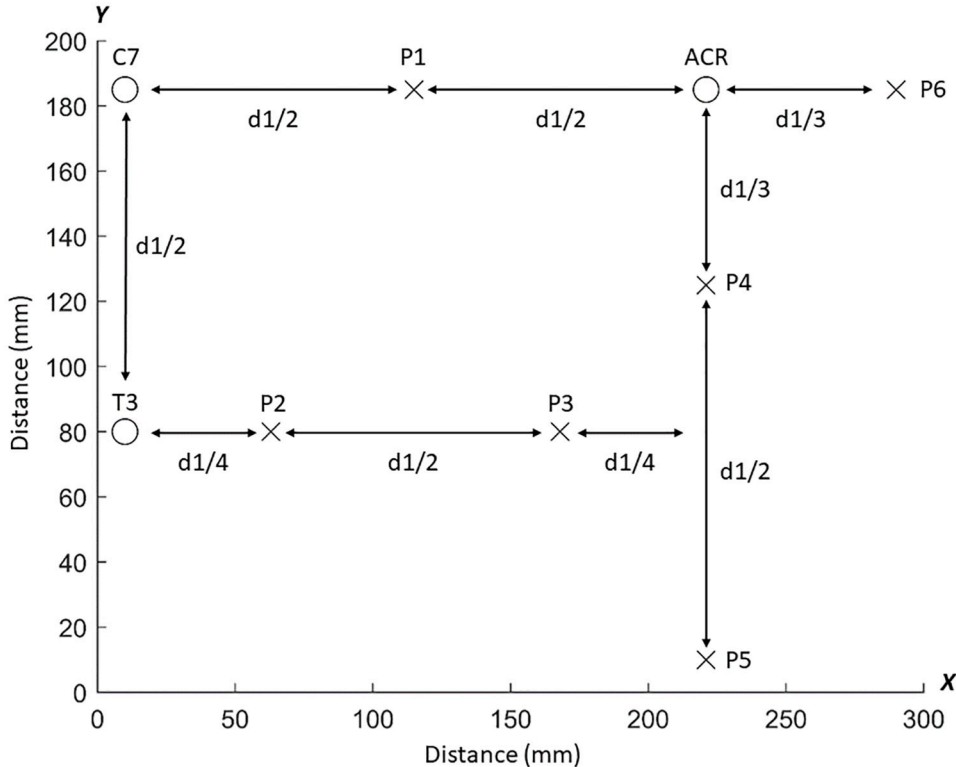

**Fig 2. Dorsal grid.** Schematic representation of the dorsal grid for pressure pain threshold (PPT) assessments. The PPTs were measured over 6 points located on the trapezius muscle (P1-P2), infraspinatus (P3), posterior deltoid (P4), latissimus dorsi (P5) and lateral deltoid (P6). d = distance between the seventh cervical vertebra (C7) and acromion (ACR) on the x-axis.

-shoulder abduction/adduction and rotation, 0˚ -elbow flexion and 0˚ -forearm supination/pronation (Fig 4B and 4C). Shoulder abduction ROM was measured as maximal shoulder abduction starting with the arm in 0˚ -shoulder flexion, abduction/adduction, 45˚ -shoulder rotation, 0˚ -elbow flexion and 90˚ -forearm supination (Fig 4D). Internal/external shoulder rotation ROM was measured as maximal internal/external rotation starting with the arm in 90˚ -shoulder abduction, 90˚ -elbow flexion, and the forearm perpendicular to the supporting surface in 0˚ of supination/pronation for supine internal/external rotation (Fig 4E and 4F).

Goniometric measurements were obtained with a goniometer by aligning the goniometer arms between bony landmarks (e.g. olecranon process and ulnar styloid of the forearm, and medial epicondyle of the humerus) and positioning the fulcrum of the goniometer above the approximate projection of the given joint center on the movement plane [28]. The assessments were performed twice over two rounds in systematic order and a third time if the measurements had a coefficient of variance $\geq$ 20%. The mean ROM values were calculated for each movement direction.

*Maximal isokinetic muscle strength (MIMS)* was measured in the shoulder flexors/extensors, horizontal extensors/flexors, abductors/adductors and internal/external rotators using an isokinetic dynamometer (Humac Norm, model 770, Computer Sports Medicine Inc., Stoughton, USA). Participants were positioned for each assessment according to the manual with stabilizing straps positioned over the hips and chest. The anatomical axis of rotation of the shoulder joint was carefully aligned with the fulcrum of the dynamometer. Each participant was

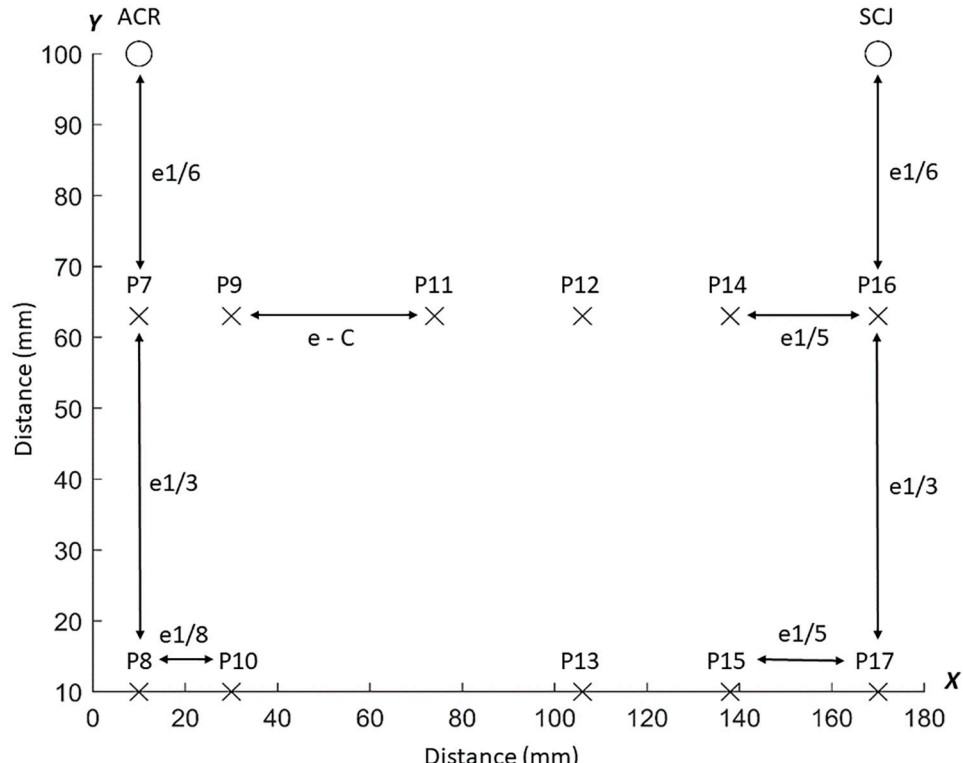

**Fig 3. Ventral grid.** Schematic representation of the ventral grid for pressure pain threshold (PPT) assessments. The PPTs were measured over 11 points located on the anterior deltoid (P7-P10) and pectoralis major (P11-P17). e = distance between the sternoclavicular joint (SCJ) and acromion (ACR) on the x-axis. C = the summed distance between P11 and P12, P12 and P14, and P14, and P16 on the x-axis.

familiarized with the protocol during FS, and a brief general warm up (approx. 10-minutes) of various stretching exercises for the prime movers was conducted prior to testing. Isokinetic strength testing was performed at a speed of 60°/s through the active ROM previously measured for each movement direction in line with Brown & Weir [43].

A series of 10 consecutive contractions with submaximal progressive effort followed by a series of five consecutive contractions at maximal effort was performed for each muscle group with a 2-minute rest period between series. Rest between strength measures for different movement patterns consisted of the time required for readjustment of the dynamometer (approximately 5 minutes). In agreement with Brown & Weir [43] the first repetition of each maximal trial was discarded and mean peak torque at a fixed position was calculated for the remaining four. Peak torque was normalized to FFM and is expressed as Nm/kg FFM [44]. The fixed joint position for each movement pattern was determined by identifying the participant that demonstrated the lowest ROM in each pattern and calculating the joint angle corresponding to midrange. Gravity correction was performed for each participant prior to the series of assessments.

## Statistical analysis

Intra-class correlation coefficient ($ICC_{2,1}$ for absolute agreement) was computed for all measures to assess the relative variability and interpreted according to Landis and Koch in which

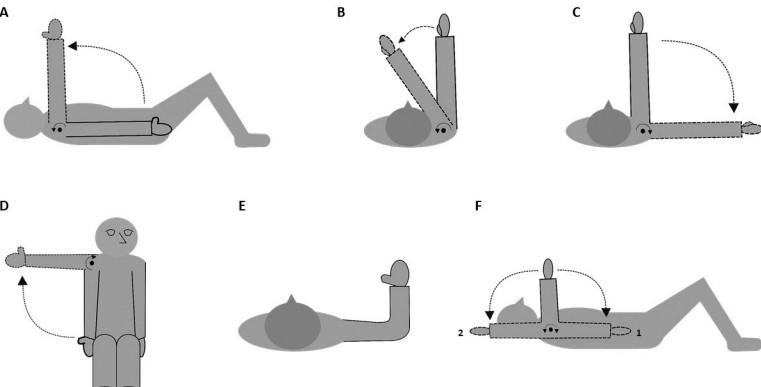

**Fig 4. Active range of motion positions.** Schematic movement directions measured for active range of motion (ROM) on the affected side; shoulder flexion (A), horizontal shoulder flexion/extension (B and C), seated shoulder abduction (D) and internal/external shoulder rotation (F, 1 (internal) and 2 (external)) performed with 90˚ shoulder abduction and elbow flexion. Starting positions are marked in bold with arrows referencing movement direction (large) and axis of rotation (small) of the arm and shoulder.

an ICC between 0.00 and 0.20 is considered "poor", 0.21–0.40 is "fair", 0.41–0.60 is "moderate", 0.61–0.80 is "substantial", and 0.81–1.00 is "almost perfect" [45]. Furthermore, Bland-Altman plots were constructed to evaluate heteroscedasticity of the measurements and detect any systematic bias in the mean difference between sessions for PPT, ROM and MIMS. Bias exists when the second set of measurements yield a different mean and heteroscedasticity occurs when the mean difference between measurements changes as their average changes [46]. Absolute reliability of each measure was estimated by computing the standard error of measurement (SEM). Using the standard deviation of the measurements from participants in both sessions (SD), SEM was calculated with the following formula:

$$SEM = SD\sqrt{1 - ICC}$$

SEM was further utilized to calculate minimal detectable chance (MDC), which represents the minimal value for which a difference can be considered as real. The formula for MDC is the following:

$$MDC = SEM \times 1.96 \times \sqrt{2}$$

A 2-way repeated measures analysis of variance (ANOVA) was performed to investigate potential differences between sessions. PPT was used as dependent factor with anatomical location (P1-17) and experimental session (ES1 & ES2) as independent factors. Similarly, active ROM and MIMS were used as dependent factors with movement direction (1–6 & 1–8) and experimental session (ES1 & ES2) as independent factors. All post hoc analyses were performed with Bonferroni correction for multiple comparisons. The Shapiro-Wilks test of normality was applied to test the assumption of normal distribution and Mauchly's test of sphericity was used to test for equality of variance in the differences between levels. If the assumption of sphericity was violated a Greenhouse-Geisser correction was applied. A P value less than 0.05 was considered statistically significant. Differences are expressed as mean (confidence interval (CI) 95%).

**Table 1. Perceived pain characteristics of the 21 women participating in this study.**

| Characteristic: | Cases (N = 21) |
|---|---|
| **Peak pain intensity, mean (CI: 95%)** | 7.2 (6.6;7.9) |
| **Pain severity, No. (%)** | |
| Light | 0 (0) |
| Moderate | 10 (48) |
| Severe | 11 (52) |
| **Location, No. (%)** | |
| Chest | 17 (81) |
| Shoulder | 8 (38) |
| Axilla | 8 (38) |
| Arm | 4 (19) |
| Side of body | 8 (38) |
| More than 1 location | 11 (52) |
| **Time with pain, mean (CI: 95%), months** | 58.9 (42.9;75) |
| **Use of pain relieving medicine, No. (%)** | 7 (33) |
| **Pain considered biggest problem post treatment, No. (%)** | 10 (48) |

Abbreviations: 95% confidence interval, CI:95%.

## Results

### Participants

Pain intensity, severity and painful locations of the 21 participants are reported in Table 1. Participants (mean (CI 95%); age: 57.4 (54;60.8) years, height 167.9 (165.5;170.2) cm, body mass index: 27.6 (25.3;29.8) kg/m$^2$) had experienced pain for a mean (95% CI) period of 58.9 (42.9;75) months prior to participating in the study. See supplemental information for detailed cohort description (S2 Appendix).

### Pressure pain threshold

Relative and absolute reliability of PPT is reported in Table 2. The ICCs of the 17 anatomical locations on the dorsal and ventral shoulder regions were "almost perfect" [45], with ICC values ranging from 0.88–0.97. SEM values ranged from 12.0 to 28.2 kPa, while MDC ranged from 33.2 to 78.2 kPa. The ICC of tibialis anterior was 0.91 with an SEM and MDC of 27.0 and 74.8 kPa respectively. The analysis of the Bland-Altman plot showed no apparent systematic bias in the data with zero included in the 95% confidence interval. Heteroscedasticity was absent indicating a lack of systematic errors during the PPT measurements. The upper-lower limits of agreement (LOA) ranged from 46.5 to 117.4 kPa and -119.9 to -45.3 kPa with a bias ranging from -13.2 to 19.7 kPa (Fig 5). See supplemental information for individual Bland-Altman plots (S3 Appendix).

The PPT maps showed that the BCS were most sensitive at the inferior portion of the pectoralis major on the ventral region, and least sensitive at the posterior deltoid on the dorsal region (Fig 6). The 2-way RM ANOVA revealed no significant interaction effect between *Location* and *Session*, *(F (2,16) = 0.889, p = 0.518)*. There was a significant main effect of *location*, *(F (2,16) = 14.211, p < 0.01)* but not of session *(F (2,1) = 0.264, p = 0.613)*. See supplemental information for pairwise comparisons among anatomical locations (S4 Appendix).

**Table 2. Test-retest reliability of each pressure pain threshold location in breast cancer survivors (N = 21).**

| PPT locations (kPa) | Mean ES 1 (95% CI) | Mean ES 2 (95% CI) | $ICC_{2,1}$ (95% CI) | SEM[1] | MDC[2] |
|---|---|---|---|---|---|
| Point 1 | 164.2 (130.0;198.3) | 152.7 (123.5;181.8) | 0.92 (0.81;0.97) | 19.4 | 53.8 |
| Point 2 | 178.1 (135.6;220.6) | 176.7 (128.1;225.2) | 0.94 (0.86;0.98) | 24.0 | 66.6 |
| Point 3 | 168.6 (135.4;201.8) | 157.8 (122.8;192.7) | 0.91 (0.79;0.96) | 22.0 | 61.0 |
| Point 4 | 194.7 (144.7;244.6) | 182.7 (139.2;226.1) | 0.92 (0.81;0.97) | 28.2 | 78.2 |
| Point 5 | 151.3 (117.6;185.1) | 159.5 (120.9;198.1) | 0.89 (0.73;0.96) | 26.2 | 72.7 |
| Point 6 | 159.3 (119.9;198.7) | 146.1 (114.5;177.7) | 0.90 (0.75;0.96) | 25.0 | 69.2 |
| **Mean.Dors** | **169.4 (130.4;208.4)** | **162.6 (124.6;200.5)** | **0.95 (0.88;0.98)** | **18.9** | **52.4** |
| Point 7 | 139.4 (115.4;163.5) | 145.7 (116.3;175.1) | 0.90 (0.74;0.96) | 18.9 | 52.4 |
| Point 8 | 126.7 (103.9;149.4) | 125.9 (102.0;149.8) | 0.94 (0.86;0.98) | 12.0 | 33.2 |
| Point 9 | 131.1 (106.5;155.7) | 132.0 (106.8;157.2) | 0.88 (0.71;0.95) | 18.6 | 51.4 |
| Point 10 | 131.9 (104.1;159.7) | 134.5 (103.4;165.5) | 0.89 (0.71;0.95) | 22.3 | 61.8 |
| Point 11 | 125.6 (92.1;159.0) | 125.2 (90.2;160.1) | 0.94 (0.85;0.98) | 18.5 | 51.2 |
| Point 12 | 114.5 (78.2;150.9) | 108.7 (82.5;135.0) | 0.89 (0.73;0.96) | 22.7 | 63.0 |
| Point 13 | 109.0 (70.6;147.3) | 97.0 (65.2;128.9) | 0.95 (0.89;0.98) | 16.5 | 45.6 |
| Point 14 | 110.6 (88.6;132.7) | 116.2 (86.7;145.8) | 0.89 (0.73;0.96) | 18.9 | 52.3 |
| Point 15 | 93.6 (73.8;113.3) | 96.7 (71.9;121.6) | 0.94 (0.84;0.97) | 12.4 | 34.5 |
| Point 16 | 112.2 (85.0;139.4) | 115.3 (88.3;142.4) | 0.93 (0.84;0.97) | 15.1 | 41.9 |
| Point 17 | 99.4 (74.5;124.3) | 100.1 (74.3;125.9) | 0.94 (0.86;0.98) | 13.0 | 36.1 |
| **Mean.Vent** | **117.6 (89.5;145.7)** | **117.9 (89.3;146.6)** | **0.98 (0.94;0.99)** | **9.4** | **26.2** |

Abbreviations: Pressure pain threshold: PPT; Experimental session: ES; Intra-class correlation coefficient: ICC; Confidence interval: CI; Standard error of measurement: SEM; Minimum detectable change: MDC; Mean dorsal shoulder: Mean.Dors; Mean ventral shoulder: Mean. Vent.

[1] using pooled SD.

[2] at the 95% confidence interval.

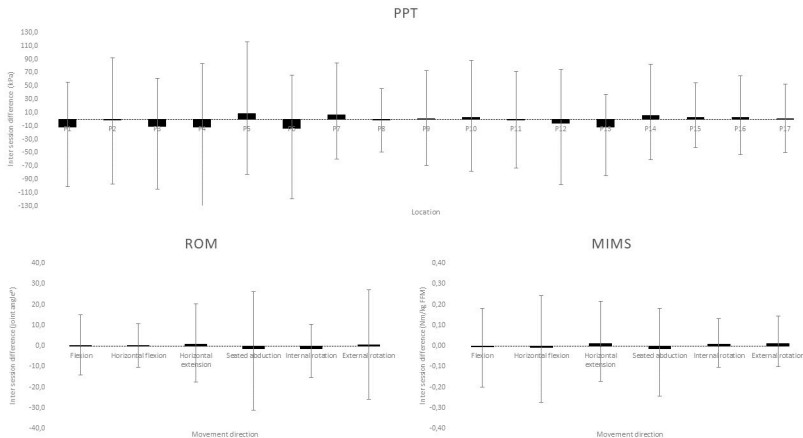

**Fig 5. Bland-Altman summary.** Summary of the Bland-Altman analysis plotted for the differences in pressures pain threshold (PPT), active range of motion (ROM) and maximal isokinetic muscle strength (MIMS) from the affected side between experimental session 1 and 2. The black boxes represent the bias and the grey error bars represents the upper-lower limits of agreement for each outcome. See supplemental information for full visualization of Bland-Altman plots (S3, S5 and S6 Appendices).

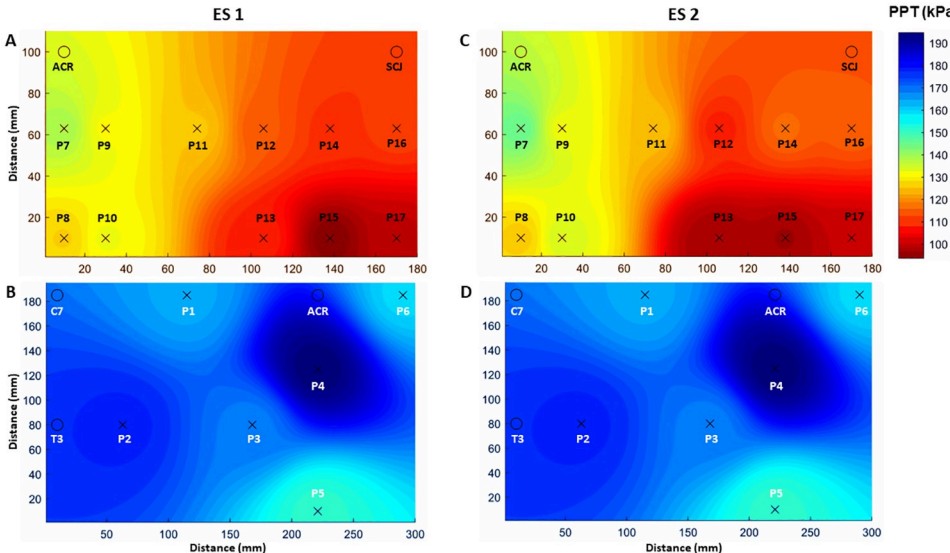

**Fig 6. Pressure pain threshold maps.** Mean pressure pain threshold (PPT) maps from the eleven anatomical locations on the ventral shoulder region, and six locations on the dorsal shoulder region of the affected side at experimental session 1 (A and B) and 2 (C and D). There were no significant difference between sessions for all locations (p> 0.05).

## Active range of motion

Relative and absolute reliability of active ROM is reported in Table 3. The ICCs of the six movement directions ranged from "substantial" to "almost perfect", with ICC values ranging from 0.66–0.97. SEM values ranged from 3.0 to 7.5˚, while MDC ranged from 8.4 to 20.8˚. The analysis of the Bland-Altman plot showed no apparent systematic bias in the data with zero included in the 95% confidence interval. Heteroscedasticity was absent indicating a lack of systematic errors during the active ROM measurements. The upper-lower limits of agreement (LOA) ranged from 10.6 to 27.4˚ and -29.8 to -10.5˚ with a bias ranging from -1.2 to 0.7˚ (Fig 5). See supplemental information for individual Bland-Altman plots (S5 Appendix).

The 2-way RM ANOVA revealed no significant interaction effect between *Direction* and *Session*, $(F_{(2,5)} = 0.168, p = 0.930)$. There was a significant main effect of *Direction*, $(F_{(2,5)} = $

**Table 3. Relative and absolute reliability of active shoulder range of motion in breast cancer survivors (N = 21).**

| Active shoulder range of motion (°) | Mean ES 1 (95% CI) | Mean ES 2 (95% CI) | ICC$_{2,1}$ (95% CI) | SEM[1] | MDC[2] |
|---|---|---|---|---|---|
| **Flexion** | 158.5 (151.3;165.7) | 158.7 (150.9;166.5) | 0.95 (0.87;0.98) | 3.7 | 10.2 |
| **Hor. Flexion** | 21.5 (19.5;23.4) | 21.5 (18.8;24.3) | 0.66 (0.15;0.86) | 3.0 | 8.4 |
| **Hor. Extension** | 97.5 (90.1;104.8) | 98.2 (90.8;105.6) | 0.91 (0.77;0.96) | 4.9 | 13.6 |
| **Abduction** | 144.2 (134.8;153.6) | 143.0 (133.0;153.0) | 0.87 (0.69;0.95) | 7.5 | 20.8 |
| **Internal rotation** | 43.2 (37.2;49.3) | 42.0 (36.5;47.5) | 0.93 (0.83;0.97) | 3.3 | 9.2 |
| **External rotation** | 72.4 (65.8;79.1) | 72.8 (65.4;80.2) | 0.78 (0.44;0.91) | 7.2 | 20.1 |

Abbreviations: Range of Motion: ROM; Joint angle: ˚; Intra-class correlation coefficient: ICC; Confidence interval: CI; Standard error of measurement: SEM; Minimum detectable change: MDC.

[1] using pooled SD.

[2] at the 95% confidence interval.

**Table 4. Relative and absolute reliability of maximum isokinetic muscle strength in breast cancer survivors (N = 21).**

| *Maximal isokinetic muscle strength* (Nm/kg FFM) | *Mean ES 1 (95% CI)* | *Mean ES 2 (95% CI)* | *ICC$_{2,1}$ (95% CI)* | SEM[1] | MDC[2] |
|---|---|---|---|---|---|
| **Extension (63˚)** | 0.47 (0.44;0.50) | 0.47 (0.42;0.52) | 0.62 (0.02;0.85) | 0.06 | 0.15 |
| **Flexion (63˚)** | 0.53 (0.48;0.58) | 0.53 (0.47;0.58) | 0.78 (0.43;0.91) | 0.05 | 0.14 |
| **Hor. extension (33˚)** | 0.43 (0.37;0.49) | 0.44 (0.37;0.51) | 0.85 (0.62;0.94) | 0.05 | 0.14 |
| **Hor. flexion (8˚)** | 0.56 (0.48;0.63) | 0.55 (0.47;0.63) | 0.83 (0.58;0.93) | 0.07 | 0.19 |
| **Abduction (58˚)** | 0.55 (0.49;0.61) | 0.53 (0.47;0.6) | 0.79 (0.47;0.92) | 0.06 | 0.16 |
| **Adduction (58˚)** | 0.40 (0.36;0.45) | 0.41 (0.37;0.46) | 0.77 (0.42;0.91) | 0.05 | 0.13 |
| **Int. rotation (11˚)** | 0.25 (0.22;0.29) | 0.26 (0.23;0.29) | 0.79 (0.48;0.92) | 0.03 | 0.09 |
| **Ext. rotation (23˚)** | 0.25 (0.21;0.28) | 0.26 (0.22;0.3) | 0.83 (0.58;0.93) | 0.03 | 0.09 |

Abbreviations: Joint angle from anatomical zero: ˚; Newton meter: Nm; Fat Free Mass: FFM; Intra-class correlation coefficient: ICC; Confidence interval: CI; Standard error of measurement: SEM; Minimum detectable change: MDC.

[1] using pooled SD.

[2] at the 95% confidence interval.

444.298, p < 0.01) but not of session (F (2,1) = 0.020, p = 0.888). See supplemental information for pairwise comparisons among movement directions (S4 Appendix).

## Maximal isokinetic muscle strength

Relative and absolute reliability of MIMS is reported in Table 4. The ICCs of the eight movement directions ranged from "substantial" to "almost perfect", with ICC values ranging from 0.62–0.92. SEM values ranged from 0.03 to 0.07 Nm/Kg FFM, while MDC ranged from 0.09 to 0.19 Nm/kg FFM. The analysis of the Bland-Altman plot showed no apparent systematic bias in the data with zero included in the 95% confidence interval. Heteroscedasticity was absent indicating a lack of systematic errors during the MIMS measurements. The upper-lower limits of agreement (LOA) ranged from 0.15 to 0.25 Nm/kg FFM and -0.27 to -0.15 Nm/kg FFM with a bias ranging from -0.02 to 0.01 Nm/kg FFM (Fig 5). See supplemental information for individual Bland-Altman plots (S6 Appendix).

The 2-way RM ANOVA revealed no significant interaction effect between *Direction* and *Session*, (F (2,7) = 0.290, p = 0.894). There was a significant main effect of *Direction*, (F (2,7) = 53.072, p < 0.01) but not of session (F (2,1) = 0.022, p = 0.882). See supplemental information for pairwise comparisons among movement directions (S4 Appendix).

## Discussion

The results of this study indicate that PPTs, active ROM and MIMS of the affected shoulder can be measured reliably across days in BCS with pain after treatment. The assessments of PPTs, active ROM and MIMS in the upper limb are of particular interest since both surgical and adjuvant treatment for breast cancer can affect these measures. These assessments provide important information of mechanical pain sensitivity and shoulder function in terms of movement and strength. Hence, the relative and absolute reliability of PPT, ROM and MIMS suggests that these measures can be used to evaluate the effects of an intervention on shoulder function and pain sensitivity in BCS.

The PPT values and distribution observed in this study are similar to those reported by Caro-Morán et al. for BCS with neck-shoulder pain approximately 20 months after treatment for breast cancer [15]. It should be noted that the participants of the present study reported pain in the chest, shoulder, axilla, arm and/or side of body approximately 70 months post treatment. This may indicate that sensitization of the group III and group IV afferents and

thus, mechanical hyperalgesia is a long lasting characteristic of BCS with pain. Interestingly, we found spatial differences within and between ventral and dorsal region with the chest area of the ventral region being the most sensitive. This area is directly affected by surgery and adjuvant radiotherapy, and thus more susceptible to nerve damage which may explain the observed hyperalgesia [7], [10].

The active ROM and MIMS values performed by the BCS were generally lower compared with asymptomatic adults [47], [48]. More specifically, the limited ROM during shoulder abduction and external rotation highlight long lasting shoulder impairments from breast cancer treatment. Notably, we also observed lower ROM and MIMS than previously reported among BCS [26], [49–54]. These differences may be due to the methodology used to assess ROM and MIMS and to the time elapsed since cancer treatment among these studies. For example, several previous studies have only provided vague descriptions of their procedure for measuring ROM and do not specify if ROM was measured actively or passively [49–53]. Considering that passive ROM is subject to measurement errors due to the fact that the stretching of soft tissues at the limits of motion depends on the force applied to the limb [55], this could explain the higher ROM values. Furthermore, this study was performed approximately 70 months post treatment compared with approximately 1–2 months [26], [49], 15 months [50], [51], 24 months [53] and 32 months [52]. This is important, because pain and fear of complications are known sources of reduced exercise activity and use of the affected limb in BCS [56]. Disuse is a source of gradual reductions in muscular strength [57] and joint motion [58], which would likely be more pronounced over time. Collectively these aspects provide possible explanations for the lower ROM and MIMS observed in the current study.

According to the GRRAS guidelines [36], reliability studies should report both relative and absolute reliability of the outcome measures. Hence, we computed ICCs and SEMs as indices of relative and absolute reliability respectively, along with MDC that served as an indicator for sensitivity to change. The ICCs in this study showed high relative reliability and were in agreement with previous studies in other populations [25], [31], [47]. ICCs showed almost perfect reliability for the PPTs and substantial to almost perfect reliability for both ROM and MIMS. Similar ICCs have been reported for PPTs in both the upper- and lower extremities of both asymptomatic and patient populations [29], [37], [59–61].

Absolute reliability, reflected in the SEM and MDC values, is difficult to compare with previous studies, as these parameters are specific to the studied population and body region. The present study is the first study to report absolute reliability for PPT, ROM and MIMS in BCS. For example, SEM for PPTs in this study ranged from 9.4–18.9 kPa for respectively the ventral and dorsal shoulder region, compared with 31–74 kPa reported for the knee region in patients with knee osteoarthritis [31]. SEM for ROM ranged from 3˚-7.5˚ which is similar to healthy individuals [47], whereas SEM for MIMS cannot be accurately compared with previous studies due to the absence of a normalization procedure in these investigations [25], [32], [48]. Body size is a known confounder of muscle strength testing and should be taken into account when evaluating muscular strength [44]. Finally, the MDCs found in this study may provide a frame of reference for evaluating the effects of an intervention on mechanical pain sensitivity and shoulder function, and hence should be reported in conjunction with significant findings [34]. However, some studies have recommended to use the percentage of participants meeting the MDC criterion instead of using the overall group change [62], [63].

The main limitation of this study is the included heterogeneous cohort. Specifically, the participant cohort included women with a broad range for age, time since treatment, and body composition. Further, participants had received varying treatment paradigms and reported pain of different intensities in different body areas. Consequently, the results are not representative of any specific group of BCS. However, BCS are arguably a heterogeneous population in

general due to the highly individual nature of diagnosis, related treatment paradigm, long-term complications and adverse effects. Thus, the inclusion of a heterogeneous cohort provide the results with a higher level of validity and generalizability for this population.

In conclusion, the results of present study suggest that PPTs, active ROM and MIMS are reliable parameters assessing mechanical pain sensitivity and shoulder function in BCS with pain after treatment. Hence, these outcome measures can be used to evaluate the effects of e.g. training interventions in this population. In addition, significant differences anatomical locations was observed in the distribution of mechanical pain sensitivity, suggesting that the ventral shoulder region is affected more by the treatment and may benefit from exercises targeting the anterior deltoid and pectoralis major.

## Supporting information

**S1 Appendix. Guidelines for Reporting Reliability and Agreement Studies (GRRAS).**
(DOCX)

**S2 Appendix. Cohort description of the breast cancer survivors of the current study.**
(DOCX)

**S3 Appendix. Bland-Altman plots for pressure pain threshold.**
(DOCX)

**S4 Appendix. Pairwise comparisons for pressure pain threshold, active range of motion and maximal isokinetic muscle strength.**
(DOCX)

**S5 Appendix. Bland-Altman plots for active range of motion.**
(DOCX)

**S6 Appendix. Bland-Altman plots for maximal isokinetic muscle strength.**
(DOCX)

## Acknowledgments

This study was conducted in collaboration with the The Society of Cancer Survivor and Late Effects Group and the Danish Cancer Society.

## Author Contributions

**Data curation:** G. H. F. Rasmussen.

**Formal analysis:** G. H. F. Rasmussen, M. Kristiansen, M. Voigt, P. Madeleine.

**Funding acquisition:** G. H. F. Rasmussen, M. Kristiansen, M. Arroyo-Morales, M. Voigt, P. Madeleine.

**Investigation:** G. H. F. Rasmussen.

**Methodology:** G. H. F. Rasmussen.

**Project administration:** G. H. F. Rasmussen.

**Supervision:** M. Kristiansen, M. Voigt, P. Madeleine.

**Writing – original draft:** G. H. F. Rasmussen.

**Writing – review & editing:** M. Kristiansen, M. Arroyo-Morales, M. Voigt, P. Madeleine.

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
