## [Decision Letter · Decision Letter 0]

1 May 2020

PONE-D-20-08793

Absolute and relative reliability of pain sensitivity and functional outcomes of the affected shoulder among women with pain after breast cancer treatment

PLOS ONE

Dear Mr Fogh Rasmussen,

Thank you for submitting your manuscript to PLOS ONE. After careful consideration, we feel that it has merit but does not fully meet PLOS ONE’s publication criteria as it currently stands. Therefore, we invite you to submit a revised version of the manuscript that addresses the points raised during the review process.

ACADEMIC EDITOR: This manuscript is very interesting. Please, address carefully the reviewers' comments in order to improve this outstanding manuscript.

We would appreciate receiving your revised manuscript by Jun 15 2020 11:59PM. To enhance the reproducibility of your results, we recommend that if applicable you deposit your laboratory protocols in protocols.io, where a protocol can be assigned its own identifier (DOI) such that it can be cited independently in the future. For instructions see: http://journals.plos.org/plosone/s/submission-guidelines#loc-laboratory-protocols

We look forward to receiving your revised manuscript.

Kind regards,

César Calvo-Lobo, PhD, MSc, PT

Academic Editor

PLOS ONE

Journal Requirements:

Additional Editor Comments (if provided):

Thanks for this interesting manuscript. Please, address carefully the reviewers' comments.

Reviewers' comments:

Reviewer's Responses to Questions

**Comments to the Author**

1. Is the manuscript technically sound, and do the data support the conclusions?

Reviewer #1: Yes

Reviewer #2: Yes

2. Has the statistical analysis been performed appropriately and rigorously? 

Reviewer #1: Yes

Reviewer #2: Yes

3. Have the authors made all data underlying the findings in their manuscript fully available?

Reviewer #1: Yes

Reviewer #2: Yes

4. Is the manuscript presented in an intelligible fashion and written in standard English?

Reviewer #1: Yes

Reviewer #2: Yes

5. Review Comments to the Author

Reviewer #1: This is a well written manuscript but some minor issues could be imporved in order to achieve a better research paper.

Introduction section is deep enough with and adequate background.

Participants section is well described is so intresting to add some references in order to enforce the inclusion and exclusion criteria

Study design section in line 112 there is a mistake with a reference, please correct it. Also in line 140.

Statistical analysis is well conducten in order to handle the mail hypoythesis

Discussion section is well explained but sometimes is difficult to follow. Please rewrite the third paragraph in order to achieve a better understunding for readers.

Conclusions are supoorted by the shown data

Reviewer #2: I have enjoyed after reading the manuscript titled “Absolute and relative reliability of pain sensitivity and functional outcomes of the affected shoulder among women with pain after breast cancer treatment”. Authors concluded that PPTs, ROM and MIMS can be measured reliably on the affected shoulder in BCS with pain after treatment. In addition, authors offer the possibility of using these measures to assess the effectiveness of interventions in this specific population. Nice study. Please, consider the following considerations:

1. Abstract and introduction provide a clear state-of-the-art and summarize the main findings of this study.

2. The methods should include a subsection called study design, including the type of study and the followed guidelines. Please, consider to add the checklist and reference of the STARD1 criteria available at https://www.equator-network.org/reporting-guidelines/stard/

1 Bossuyt PM, Reitsma JB, Bruns DE, Gatsonis CA, Glasziou PP, Irwig L, LijmerJG Moher D, Rennie D, de Vet HCW, Kressel HY, Rifai N, Golub RM, Altman DG, Hooft L, Korevaar DA, Cohen JF, For the STARD Group. STARD 2015: An Updated List of Essential Items for Reporting Diagnostic Accuracy Studies. BMJ. 2015;351:h5527. PMID: 26511519

3. In participants’ section, the following sentence needs to be referenced in an correct way: “The PPT, active range of motion (ROM) and MIMS were measured unilaterally on the operated side in all sessions”. Please, correct it.

4. The same in the following sentence of pain sensitivity and mechanical sensitivity measures section: “Adjacent points were separated by varying distances according to their anatomical location relative to the reference points”.

5. Nice statistical analysis and results sections.

6. It would be adequate to add a brief commentary or discussion about supplemental information for individual Bland-Altman plots. It seems that heteroscedasticity was not shown and thus there were not systematic errors of measurement.

6. PLOS authors have the option to publish the peer review history of their article (what does this mean?). If published, this will include your full peer review and any attached files.

Reviewer #1: No

Reviewer #2: No

---

## [Author Response · Author response to Decision Letter 0]

15 May 2020

PONE-D-20-08793

Absolute and relative reliability of pain sensitivity and functional outcomes of the affected shoulder among women with pain after breast cancer treatment

Reviewers' comments:

Reviewer #1: 

This is a well written manuscript but some minor issues could be imporved in order to achieve a better research paper. 

Response: Thank you for your positive feedback as well as constructive and useful comments that have contributed to improve the overall readability and quality of the manuscript. Changes made in the manuscript according to your comments are highlighted in yellow. Please find the specific actions and responses to you suggestions below.

Introduction section is deep enough with and adequate background.

Response: Thank you once again for your positive feedback.

Participants section is well described is so intresting to add some references in order to enforce the inclusion and exclusion criteria.

Response: The inclusion and exclusion criteria were the ones defined in our application to the ethics committee in line with e.g. studies by Caro-Morán and colleagues. These criteria were developed based on our experience with clinical studies and on the discussion we have had with Professor Niels Kroman, breast surgeon and leading physician at the Danish Cancer Society. We have added a reference that have utilized similar inclusion and exclusion criteria on page 5, line 91-92:

“The inclusion and exclusion criteria were similar to those used by Caro-Morán & colleagues (15).” 

Study design section in line 112 there is a mistake with a reference, please correct it. Also in line 140.

Response and action: We corrected these mistakes. Please see highlighted changes on page 6, line 116 and page 7, line 144.

Statistical analysis is well conducten in order to handle the mail hypoythesis

Response: Thank you, we are grateful for your positive comments. 

Discussion section is well explained but sometimes is difficult to follow. Please rewrite the third paragraph in order to achieve a better understunding for readers.

Response and action: The 3rd paragraph of the discussion was edited to facilitate its understanding. We have reordered our argumentation starting with ROM and MIMS values and continuing with the effects of the time elapsed since treatment. Se highlighted changes below:

The active ROM and MIMS values performed by the BCS were generally lower compared with asymptomatic adults (47,48). More specifically, the limited ROM during shoulder abduction and external rotation highlight long lasting shoulder impairments from breast cancer treatment. Notably, we also observed lower ROM and MIMS than previously reported among BCS (26,49–54). These differences may be due to the methodology used to assess ROM and MIMS and to the time elapsed since cancer treatment among these studies. For example, several previous studies have only provided vague descriptions of their procedure for measuring ROM and do not specify if ROM was measured actively or passively (49–53). Considering that passive ROM is subject to measurement errors due to the fact that the stretching of soft tissues at the limits of motion depends on the force applied to the limb (55), this could explain the higher ROM values. Furthermore, this study was performed approximately 70 months post treatment compared with approximately 1-2 months (26,49), 15 months (50,51), 24 months (53) and 32 months (52). This is important, because pain and fear of complications are known sources of reduced exercise activity and use of the affected limb in BCS (56). Disuse is a source of gradual reductions in muscular strength (57) and joint motion (58), which would likely be more pronounced over time. Collectively these aspects provide possible explanations for the lower ROM and MIMS observed in the current study.

Conclusions are supoorted by the shown data.

Response: Thank you for the positive evaluation of our manuscript.

 

Reviewer #2:

I have enjoyed after reading the manuscript titled “Absolute and relative reliability of pain sensitivity and functional outcomes of the affected shoulder among women with pain after breast cancer treatment”. Authors concluded that PPTs, ROM and MIMS can be measured reliably on the affected shoulder in BCS with pain after treatment. In addition, authors offer the possibility of using these measures to assess the effectiveness of interventions in this specific population. Nice study. Please, consider the following considerations:

Response: Thank you for your positive feedback as well as constructive and useful comments that have contributed to improve the overall readability and quality of the manuscript. Changes made in the manuscript according to your comments are highlighted in green. Please find the specific actions and responses to you suggestions below.

1. Abstract and introduction provide a clear state-of-the-art and summarize the main findings of this study.

Response: Thank you, these positive comments are most appreciated.

2. The methods should include a subsection called study design, including the type of study and the followed guidelines. Please, consider to add the checklist and reference of the STARD1 criteria available at https://www.equator-network.org/reporting-guidelines/stard/

Response and action: We added a subsection entitled “Study design”, including the type of study and the followed guidelines. It is our understanding that the STARD guidelines refer more to accuracy studies than reliability studies like this one. Thus, we did not include the suggested STARD criteria. Rather, we clearly emphasized that this study was conducted in accordance with the Guidelines for Reporting Reliability and Agreement Studies (GRRAS) by adding the following sentence on page 6, lines 108-110. See also S1 appendix.

“This intra-rater reliability study was conducted in accordance with the GRRAS guidelines (37). In line with the GRRAS guidelines, we report information on sample selection, study design, and statistical analysis (S1 appendix).”

3. In participants’ section, the following sentence needs to be referenced in an correct way: “The PPT, active range of motion (ROM) and MIMS were measured unilaterally on the operated side in all sessions”. Please, correct it.

Response and action: We corrected that sentence accordingly adding that the sentence refers to figure 1. Please note that the correction is highlighted in yellow on page 6, line 116

4. The same in the following sentence of pain sensitivity and mechanical sensitivity measures section: “Adjacent points were separated by varying distances according to their anatomical location relative to the reference points”.

Response and action: We also corrected that sentence accordingly adding that the sentence refers to figure 2 and 3. Please note that the correction is highlighted in yellow on page 7, line 144.

5. Nice statistical analysis and results sections.

Response: Thank you, we appreciate your positive feedback.

6. It would be adequate to add a brief commentary or discussion about supplemental information for individual Bland-Altman plots. It seems that heteroscedasticity was not shown and thus there were not systematic errors of measurement.

Response and action: We added comments in the methods and results section related to the Bland-Altman plots (heteroscedasticity and lack of systematic errors) in order to further elaborate this part of the analysis. See highlighted changes below. 

Page 10, line 215-219:

“Bland-Altman plots were constructed to evaluate heteroscedasticity of the measurements and detect any systematic bias in the mean difference between sessions for PPT, ROM and MIMS. Bias exists when the second set of measurements yield a different mean and heteroscedasticity occurs when the mean difference between measurements changes as their average changes (46).”

Page 13, line 256-257:

“Heteroscedasticity was absent indicating a lack of systematic errors during the PPT measurements.”

Page 16, line 288-289:

“Heteroscedasticity was absent indicating a lack of systematic errors during the active ROM measurements.”

Page 17, line 307-308:

“Heteroscedasticity was absent indicating a lack of systematic errors during the MIMS measurements.”

---

## [Editor Report · Decision Letter 1]

20 May 2020

Absolute and relative reliability of pain sensitivity and functional outcomes of the affected shoulder among women with pain after breast cancer treatment

PONE-D-20-08793R1

Dear Dr. Fogh Rasmussen,

We are pleased to inform you that your manuscript has been judged scientifically suitable for publication and will be formally accepted for publication once it complies with all outstanding technical requirements.

With kind regards,

César Calvo-Lobo, PhD, MSc, PT

Academic Editor

PLOS ONE

Additional Editor Comments (optional):

Thanks for addressing all reviewers' suggestions in a correct way. 
---

## [Editor Report · Acceptance letter]

22 May 2020

PONE-D-20-08793R1 

Absolute and relative reliability of pain sensitivity and functional outcomes of the affected shoulder among women with pain after breast cancer treatment 

Dear Dr. F. Rasmussen:

I am pleased to inform you that your manuscript has been deemed suitable for publication in PLOS ONE. Congratulations! Your manuscript is now with our production department. 

With kind regards,

on behalf of

Dr. César Calvo-Lobo 

Academic Editor

PLOS ONE